# Impact of delayed type hypersensitivity arthritis on development of heart failure by aortic constriction in mice

**Theis Christian Tønnessen**[1,2,3]*, **Arne Olav Melleby**[3,4], **Ida Marie Hauge-Iversen**[3,4],
**Emil Knut Stenersen Espe**[3,4], **Mohammed Shakil Ahmed**[2], **Thor Ueland**[5],
**Espen Andre Haavardsholm**[3,6], **Sara Marie Atkinson**[7], **Espen Melum**[3,8,9,10,11],
**Håvard Attramadal**[2,3], **Ivar Sjaastad**[3,4], **Leif Erik Vinge**[1,2,3]

**1** Department of Medicine, Diakonhjemmet Hospital, Oslo, Norway, **2** Institute for Surgical Research, Oslo University Hospital, Oslo, Norway, **3** Institute of Clinical Medicine, Faculty of Medicine, University of Oslo, Oslo, Norway, **4** Institute for Experimental Medical Research and KG Jebsen Center for Cardiac Research, Oslo University Hospital and University of Oslo, Oslo, Norway, **5** Research Institute of Internal Medicine, Oslo University Hospital, Oslo, Norway, **6** Department of Rheumatology, Diakonhjemmet Hospital, Oslo, Norway, **7** Skin Research, LEO Pharma A/S, Ballerup, Denmark, **8** Norwegian PSC Research Center, Department of Transplantation Medicine, Division of Surgery, Inflammatory Diseases and Transplantation, Oslo University Hospital Rikshospitalet, Oslo, Norway, **9** Research Institute of Internal Medicine, Division of Surgery, Inflammatory Diseases and Transplantation, Oslo University Hospital, Oslo, Norway, **10** Section for Gastroenterology, Department of Transplantation Medicine, Division of Surgery, Inflammatory Diseases and Transplantation, Oslo University Hospital Rikshospitalet, Oslo, Norway, **11** Hybrid Technology Hub-Centre of Excellence, Institute of Basic Medical Sciences, Faculty of Medicine, University of Oslo, Oslo, Norway

* theischristiant@gmail.com

## Abstract

### Aims

Patients with rheumatoid arthritis (RA) have increased risk of heart failure (HF). The mechanisms and cardiac prerequisites explaining this association remain unresolved. In this study, we sought to determine the potential cardiac impact of an experimental model of RA in mice subjected to HF by constriction of the ascending aorta.

### Methods

Aorta was constricted via thoracotomy and placement of o-rings with inner diameter 0.55 mm or 0.66 mm, or sham operated. RA-like phenotype was instigated by delayed-type hypersensitivity arthritis (DTHA) two weeks after surgery and re-iterated after additional 18 days. Cardiac magnetic resonance imaging (MRI) was performed before surgery and at successive time points throughout the study. Six weeks after surgery the mice were euthanized, blood and tissue were collected, organ weights were documented, and expression levels of cardiac foetal genes were analysed. In a supplemental study, DTHA-mice were euthanized throughout 14 days after induction of arthritis, and blood was analysed for important markers and mediators of RA (SAP, TNF-α and IL-6). In order to put the latter findings into clinical context, the same molecules were analysed in serum from untreated RA patients and compared to healthy controls.

**Data Availability Statement:** All relevant data are within the paper and its Supporting information files.

**Funding:** This work was supported by a grant to LEV from The Bergesen foundation, Norway. The funders had no role in study design, data collection and analysis, decision to publish, or preparation of the manuscript. URL: https://bergesenstiftelsen.no/.

**Competing interests:** The authors have declared that no competing interests exist.

**Abbreviations:** ACTA1, actin alpha 1, skeletal muscle; CAIA, collagen antibody-induced arthritis; cDNA, complementary deoxyribonucleic acid; CFA, complete Freund's adjuvant; CRP, C-reactive protein; $C_T$, cycle threshold; CVD, cardiovascular disease; DAS, disease activity score; DTHA, delayed type hypersensitivity arthritis; ECG, electrocardiography; EDTA, ethylenediaminetetraacetic acid; EF, ejection fraction (%); EIA, enzyme immunoassay; GLS, global longitudinal strain (%); HF, heart failure; HFpEF, heart failure with preserved ejection fraction; HFrEF, heart failure with reduced ejection fraction; IL-6, interleukin 6; LV, left ventricle; LVEDV, LV end-diastolic volume; LVESV, LV end-systolic volume; mBSA, methylated bovine serum albumin; MRI, magnetic resonance imaging; mRNA, messenger ribonucleic acid; MYH7, myosin heavy chain 7; NPPA, natriuretic peptide A; NPPB, natriuretic peptide B; ORAB, o-ring aortic banding; qPCR, quantitative polymerase chain reaction; RA, rheumatoid arthritis; RNA, ribonucleic acid; rRNA, ribosomal ribonucleic acid; SAP, serum amyloid P component; SRe', peak early diastolic strain rate (1/s); SRs', peak systolic strain rate (1/s); SV, stroke volume; TNF-α, tumor necrosis factor α; TPM, tissue phase mapping.

## Results

Significant elevations of inflammatory markers were found in both patient- and murine blood. Furthermore, the DTHA model appeared clinically relevant when compared to the inflammatory responses observed in three prespecified RA severity disease states. Two distinct trajectories of cardiac dysfunction and HF development were found using the two o-ring sizes. These differences were consistent by both MRI, organ weights and cardiac foetal gene expression levels. Still, no difference within the HF groups, nor within the sham groups, could be found when DTHA was induced.

## Conclusion

DTHA mediated systemic inflammation did not cause, nor modify HF caused by aortic constriction. This indicates other prerequisites for RA-induced cardiac dysfunction.

## 1. Introduction

Both rheumatoid arthritis (RA) and cardiovascular diseases (CVD) have high global prevalence [1, 2]. Approximately 0.5–1.0% of the general population suffers from RA, with considerable variance with respect to hemisphere (more frequent in the northern parts), region (more frequent in urban areas) and sex (more frequent in females) [3], while heart failure (HF) affects approximately 2% of the general population. The risk of developing HF is age-related with a 30% life-time risk of developing HF after 55 years of age [1]. The predominant causes of HF in the western hemisphere are ischemic heart disease and hypertension [4]. However, numerous other conditions, including systemic inflammatory diseases, underlie collectively a substantial proportion of the HF cases [4].

It is well established that RA increases the risk of coronary heart disease [5]. In fact, RA is a numerically defined risk factor in current clinical CVD guidelines [6]. However, it has also been reported that RA increases the risk of HF, even after correction for other CVD risk factors and importantly, exclusion of coronary heart disease [7]. This suggests the presence of RA-specific HF risks. Epidemiological studies suggest an approximately two-fold increased risk of HF with RA [8, 9]. However, these studies do not differentiate between heart failure with reduced ejection fraction (HFrEF) versus heart failure with preserved ejection fraction (HFpEF). Short-duration, prospective studies have shown that an even higher proportion of RA patients develop cardiac dysfunction (i.e., subclinical HF). Of patients with newly diagnosed RA and initial normal heart function, 24% developed mild left ventricular (LV) diastolic dysfunction after one year [10]. However, no longer duration studies have been undertaken, and thus further cardiac deterioration (i.e., into HFrEF) cannot be excluded. Interestingly, the risk of HF is even higher amongst women and appears to increase with age [7]. Finally, data comparing HF patients with or without concomitant RA indicate a doubled mortality in the RA/HF comorbidity as compared to HF alone [11].

Knowledge of the RA-specific HF risk factors is so far limited. RA disease severity has been shown to influence the occurrence of HF [9, 12, 13], and some studies suggest that degree of systemic inflammation is the nodal factor dictating the occurrence and degree of HF [9, 12, 13]. Pharmacological adjustments of RA disease activities also seem to impact HF [12, 14]. However, none of these studies provides definite clues as to whether these findings are due to

direct cardiac effects of therapeutic intervention or indirectly through the consequence of decreased systemic inflammation.

Most of the above-mentioned studies are epidemiological. Fewer studies are prospective, and these are all small, of short duration and purely observational. Still, these studies collectively unequivocally document the association between RA and HF. However, they do not allow mechanistic insights, nor address whether RA alone is a sufficient risk factor or require pre-existing cardiac vulnerability. Addressing these highly relevant clinical questions will be exceedingly challenging in human trials and as of yet only a few experimental studies investigating the cardiac impact of experimental RA exists [15–19]. A fundamental question is whether the observed cardiac dysfunction in RA is a consequence of the same pathogenesis (i.e., autoimmunity/autoantibodies) or an indirect effect of resulting systemic inflammation. No studies as of yet have addressed this. However, cardiac dysfunction is also seen in other chronic inflammatory conditions (like inflammatory bowel disease, psoriasis arthritis, ankylosing spondylitis) where the autoimmunogenic pathogenesis is different [20–22]. Furthermore, the few experimental RA models in which cardiac function has been studied also varies in respect to pathogenesis [15–19]. Based on these clinical and experimental observations it is tempting to suggest the common denominator being systemic inflammation.

In this study, we have chosen a different experimental RA model than previously used, the delayed type hypersensitivity arthritis (DTHA) model. Using this model, we have examined its effect on normal hearts, as well as on the natural development of two different HF trajectories. Although not excluded, the regionally restricted arthritis phenotype probably allows for viewing putative cardiac findings as a consequence of arthritis-induced systemic inflammation rather than autoantibodies directly injuring the heart.

## 2. Methods

### 2.1 Ethics

The experimental study was performed in accordance with EU directive 2010/63/EU (the Norwegian Food Safety Authority, "Mattilsynet", approval FOTS ID #14385) and in agreement with the Guide for the Care and Use of Laboratory Animals (NIH publication No. 85–23, revised 2011, US). The reporting of experimental procedures and results were performed in accordance with the ARRIVE guidelines. In the supplemental study we included patients from a clinical study [23] and matched biobanked healthy controls [24]. The clinical study and the biobanked material were in compliance with the Declaration of Helsinki and approved by Regional Committees for Medical Research Ethics in South Eastern Norway (REC South East Norway, reference number 2010–744 [23], 2012–286 and 2016–1540 [24], respectively), and all patients and healthy controls provided written informed consent.

### 2.2 Mouse o-ring aortic banding (ORAB)

A schematic timeline of the experimental protocol can be viewed in S1 Fig. 10-week-old male C57BL/6J mice (Janvier labs, Le Genest-Saint-Isle, France) weighing 25.0 g (± 0.7) at the time of surgery were housed in a facility with 12-hour light/dark cycle and *ad libitum* supply of water and standard rodent chow. Mice (n = 50) were housed in cages with 5 in each cage, and were randomized to the ORAB and RA treatments irrespective of the cage. The mice were allowed to acclimatize for at least one week before surgery. Mice were then subjected to thoracic aortic constriction by ORAB surgery with ring inner diameter 0.66 mm (n = 18) or 0.55 mm (n = 18) as previously described [25]. Sham operated mice (mice undergoing the entire surgical procedure except placement of o-ring) were included as controls (n = 14). Decision of degree of ORAB was based upon the above referenced method paper anticipating two distinct

trajectories of cardiac hypertrophy/remodelling and dysfunction. In brief, mice were intubated and ventilated with a gas mixture containing 98% oxygen and 2% isoflurane. A skin incision was made on the left side of the thorax, and access to the ascending aorta was established by blunt dissection through the 3$^{rd}$ intercostal space, retraction of the costae and the thymus and removal of periaortic fat and fibrous tissue. A cut o-ring with pre-attached ligatures at each end was subsequently pulled under and around the aorta and tied. Lungs were inflated, incision was covered by the pectoral muscles and the skin closed with continuous sutures. Mice were kept on ventilation until commencing of spontaneous breathing. Mice received analgesia by subcutaneous injection of 0,2 mg/kg buprenorphine pre-operatively and at 8, 16 and 24 hours post-operatively. Additional analgesics were administered subject to animal status.

## 2.3 Induction and assessment of DTHA in ORAB-operated mice

One week after surgery mice were randomized 1:1 to DTHA or control. Arthritis was induced 14 days after ORAB. In brief, mice were intradermally injected at the base of the tail $2 \times 100$ μL of methylated bovine serum albumin (mBSA; 2.5 mg/mL in PBS; Sigma-Aldrich, St. Louis, MO, USA), emulsified 1:1 with complete Freund's adjuvant (CFA; Difco, Detroit, MI, USA) containing 5 mg/mL Mycobacterium butyricum. Four days after intradermal injection mice were intravenously injected 200 μL of anti-collagen II antibodies (5-clone cocktail; 5 mg/mL; Chondrex, Redmond, WA, USA). Finally, after additional three days, arthritis was induced by subcutaneous injection of 20 μL of mBSA (10 mg/mL) in the right hind leg footpad [26]. Sham-RA mice underwent the same injection procedures, but with vehicle. Arthritis phenotype was assessed serially by standardized photography of hind leg (sagittal projection) towards a ruler. Photographs were subsequently used to measure hind leg diameter using ImageJ. As the DTHA model is of transient nature, arthritis was re-established by a subsequent i.v. injection of anti-collagen II antibodies at day 15 after initial induction of arthritis, and subsequent s.c. injection of mBSA at day 18.

## 2.4 Cardiac magnetic resonance imaging

Cardiac magnetic resonance imaging (MRI) was performed on 7 mice from each of the six intervention groups (ORAB 0.55 and 0.66, and sham, with or without RA) at day 14, 28 and 42 after ORAB surgery. Mice were randomly picked from each group for MRI scanning before entering the arthritis-inducing protocol. In addition, a randomized selection of mice (n = 12) were examined by MRI one week before ORAB surgery in order to generate an unpaired baseline.

MRI scanning was performed as previously described [25, 27]. In brief, the 9.4 T MRI system (Agilent, Palo Alto, CA, USA) with a 35 mm quadrature-driven birdcage RF coil (Rapid Biomedical, Rimpar, Germany) dedicated to mouse imaging was used. Anaesthesia given by mask during the MRI examination was 1–2.5% isoflurane and $O_2$. Mice were positioned prone in a dedicated animal bed. Body temperature, respiration and ECG were continuously monitored during examination. Temperature was automatically maintained using heated air. The level of anaesthesia was manually adjusted throughout the examination to ensure stable respiration and heart rate. Cine-MRI was acquired in using a stack of short-axis slices covering the LV, and TPM-MRI was acquired in four-chamber long-axis orientation. All MRI data was ECG- and respiratory-gated. Central imaging parameters were repetition time 5.0 ms for cine-MRI and 4.48 ms for tissue phase mapping (TPM)-MRI, acquisition matrix 128 x 128 and 96 x 96 for cine- and TPM-MRI, respectively. Field-of-view was 25 x 25 mm and slice thickness 1 mm for both cine- and TPM-MRI.

**2.4.1 MRI post-processing.** Segmentation of the cine-MRI was conducted using Matlab (The MathWorks Inc., USA), where endocardium and epicardium were detected in a semiautomatic manner. Briefly, the semiautomatic method took the centrum of the left ventricle as user input and converted the images into polar coordinates. The contrast between blood and myocardium were automatically detected and were used to generate endo- and epicardial masks. These masks were then manually validated and adjusted, if necessary, by a single examiner. From the cine-MRI data, left ventricular (LV) mass, LV end-diastolic volume (LVEDV), LV end-systolic volume (LVESV), stroke volume (SV), and LV ejection fraction (LVEF) were calculated.

Analysis of the TPM-MRI data was conducted using Matlab (The MathWorks Inc., USA) as previously described in detail [27]. TPM-MRI was used to measure global longitudinal strain (GLS, %), peak early diastolic strain rate (SRe', 1/s) and peak systolic strain rate (SRs', 1/s) in the septum and the LV free wall, and global. SRs' and SRe' were measured from the peaks in the strain rate curve gained from TPM using Matlab (The MathWorks Inc., USA). All MRI-analysis was conducted by staff blind to the treatment of the animals.

## 2.5 Tissue- and blood sampling

Mice were euthanized deeply anesthetized in a mixture of oxygen and 3–4% isoflurane by extraction of the heart. Organs were harvested and weighed before the heart were cut at the transverse midventricular plane. The apical part of the LV was snap frozen in liquid nitrogen and stored at -80˚C until use. In the supplemental study, mice were euthanized by cutting the carotid artery as described in 2.7.

## 2.6 Isolation of RNA and analysis of myocardial gene expression

Total RNA was isolated from myocardial tissue samples by homogenization in chaotropic salts and subsequent ion exchange chromatography with the RNeasy system (Qiagen) and kept in RNase-free water containing RNase inhibitor (RNasin; Promega) at $-70$˚C until further use. For real-time qPCR, total RNA was reverse transcribed with the TaqMan Reverse Transcription Reagents Kit, and subsequently real-time qPCR of each sample was run in triplicate with TaqMan Pre-Developed Assay Reagents and the StepOnePlus™ Real-Time PCR System and software (Applied Biosystems, Foster City, CA) according to the manufacturer's instructions. A standard curve was obtained by amplifications of cDNA obtained from serial dilutions of myocardial RNA. Atrial natriuretic peptide (*NPPA*), brain natriuretic peptide (*NPPB*), β-myosin heavy chain (*MYH7*), and skeletal alpha actin (*ACTA1*) were measured. For all specific mRNA amplified, linear inverse correlations were observed between amount of mRNA and $C_T$ value (number of cycles at threshold lines). Gene expression is presented relative to the levels of 18S rRNA as the housekeeping gene.

## 2.7 Assessment of degree of clinically relevant systemic inflammatory responses in the DTHA model

To validate the clinical translational value of the inflammatory responses in the circulation of the murine DTHA model, we conducted a supplemental time course study where mice subjected to DTHA or control protocol were euthanized sequentially for 14 days. Mice were anesthetized with a gas mixture containing 96–97% oxygen and 3–4% isoflurane, and a midline incision was made from the mandibulae towards the sternum. With blunt dissection the carotid artery was identified and subsequently cut. 50 μL 0.5 M EDTA was deposited within the incision followed by extraction of blood into a 100cc syringe. Blood was transferred to a pre-chilled 1.5 mL centrifuge tube and subsequently centrifuged at 4˚C for 20 minutes at 2000 × g.

Plasma was aliquoted and snap frozen in liquid nitrogen and stored at -80°C until use. Mouse plasma was subsequently analysed for SAP using enzyme immunoassay (EIA) (Hycult Biotech, Uden, Netherlands), and IL-6 and TNF-α were analysed using MSD cytokine assay (Meso Scale Discovery, MSD, Rockville, MD, USA). To validate the significance of the relative increments of key inflammatory parameters in the DTHA model we analysed CRP, IL-6 and TNF-α in plasma from patients with untreated seropositive RA. The patient characteristics have previously been published [23]. This study, however, did not include age- and sex-matched healthy volunteers, a necessary prerequisite for assessing the relative increments of inflammatory mediators needed for comparing the human and murine data. Such controls were recruited by the Norwegian PSC Research Center biobank after providing written informed consent [24]. Patients were drawn from the ARCTIC biobank according to disease severity assessed by Disease Activity Score (DAS), and thus categorized in a mild, moderate and severe disease group. Human plasma was analysed using MSD cytokine assay for IL-6 and TNF-α, and EIA was used for CRP (RnD systems, Stillwater, MN). Intraassay coefficients of variation were <10%.

## 2.8 Statistics

Data and figures are presented as means with standard error of the mean. The statistical calculations were made in GraphPad Prism 9 statistical package (GraphPad Software, La Jolla, CA). For comparisons of two groups, we used unpaired *t*-test. For non-repeated measures with more than two groups, we used one-way ANOVA with Bonferroni's multiple comparisons test. For repeated measures, we used two-way ANOVA with Bonferroni's multiple comparisons test. To identify outliers, we used the ROUT method with a Q value of 1%. Statistical significance was defined as $P < 0.05$.

## 2.9 Evaluation of animal status

We evaluated the animal status daily with a score sheet with prespecified criteria, based on posture, fur, grooming, weight loss, and clinical features of limited gait. Based on the score we assessed the need for pain relief and closer monitoring, or if euthanization was necessary. In addition, a footpad swelling > 4 mm compared to baseline or a wound at injection site > 8 mm required immediate euthanization.

## 2.10 Calculating group sizes

Group sizes were estimated using power analysis. We chose a significance level of 0.05 and power of 0.8 resulting in constant C = 7.85 according to Guidelines for the Care and Use of Mammals in Neuroscience and Behavioural Research [28]. Based on current available data [15], we hypothesised a mean difference (D) in heart function of 15% between RA+ and RA-, measured by ejection fraction (EF, %) on MRI, and a standard deviation (SD) of 10%. The number of required animals is given by:

n = 1+2C $(SD/D)^2$. Hence n = 1+15.7$(0.10/0.15)^2$ = 8. As we expected a slightly higher standard deviation in the ORAB-groups than in sham animals, we chose to have only 7 animals in the sham groups, and more animals in the ORAB-groups.

## 2.11 Criteria for exclusion of animals

We did have preset criteria for exclusion of animals according to failure of surgery, faulty induction of arthritis, animal welfare (see Methods section 2.9 above) and so forth. However, none of these were met (i.e., none were excluded).

### 2.12 Blinding

TCT was the only investigator aware of the group allocation during the conduct of the experiment, and blinding was initiated before experimentation and analysis involving TCT.

## 3. Results

### 3.1 Inflammatory responses in the DTHA model

**3.1.1 Establishment and assessment of the DTHA model.** Arthritis phenotype was seen in 100% of the animals in accordance with the original method paper [26]. The phenotype was macroscopically restricted to the right hind leg (where the antigen was s.c. injected) and the variance was low (see Fig 1, panel A). Significant swelling was observed from day 1 through day 8. Also, as seen in Fig 2, panel D-F, we observed a robust, although transient increase of clinically relevant inflammatory mediators in circulation. Figures displaying absolute data are shown in S2 Fig, panel D-F. One mouse subjected to the ORAB 0.66 DTHA group died at day 12 after establishment of DTHA for unknown reasons. There were no other indications that mice subjected to DTHA failed to thrive as they had a similar gain of body weight as compared to the control mice (Fig 3A).

**3.1.2 Validation of systemic inflammatory responses and the putative clinical relevance of the DTHA model.** As this study aimed to experimentally address the putative clinical negative association between RA and cardiac function, establishing the clinical relevance of our RA model was highly important. Clinical relevance in this aspect reflects the model's pathophysiological relevance, type of immune response as well as the magnitude and duration of the inflammatory response. As previously described, the DTHA model displays a pathogenesis at least partly resembling the human counterpart [26, 29]. The experimental arthritis phenotype is triggered by an unspecific antigen (mBSA, this resembling a neoantigen) and it generates a familiar local inflammation with systemic "spillover". Furthermore, the DTHA-induced arthritis can be abrogated by the most common human anti-inflammatory biologic drugs [29]. Although not studied, the DTHA model does probably not convey "autoantibody"-mediated actions on other tissues than the neoantigen-injected hind-limb. Thus, the DTHA model may represent a more isolated aspect (systemic inflammation) of RA-mediated cardiac effects, an aspect which is not as easily deciphered in other experimental RA-models.

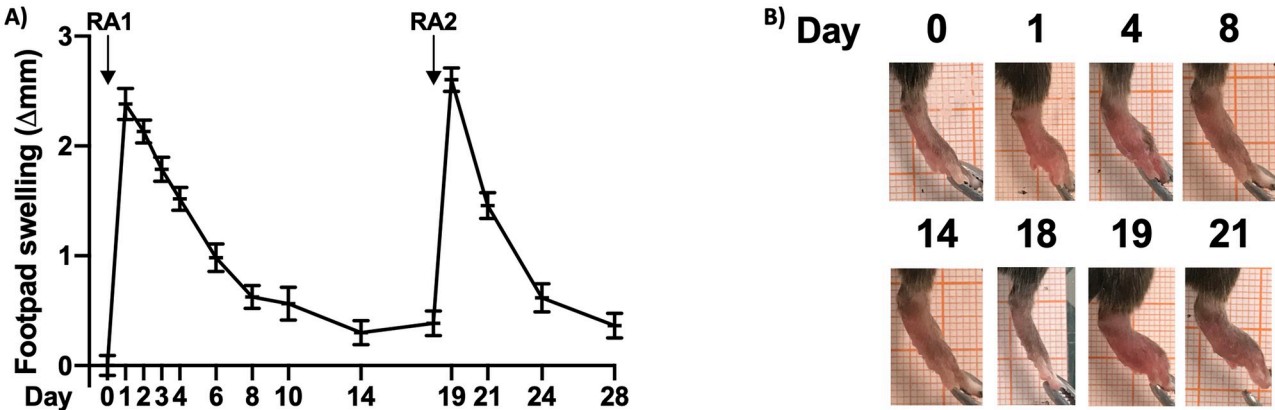

**Fig 1. Assessment of footpad swelling in the DTHA model.** *Panel A*: Footpad swelling from day 0 through day 28 after arthritis induction by s.c. injection of mBSA (RA1). At day 15, mice received an additional i.v. injection of anti-collagen II antibodies before an additional injection with mBSA (day 18, RA2), re-establishing arthritis. Figure displays increase of sagittal diameter in mm from baseline. Data is shown as mean ± SEM (n = 25). mBSA, methylated bovine serum albumin. *Panel B*: Representative photographs displaying temporal paw swelling.

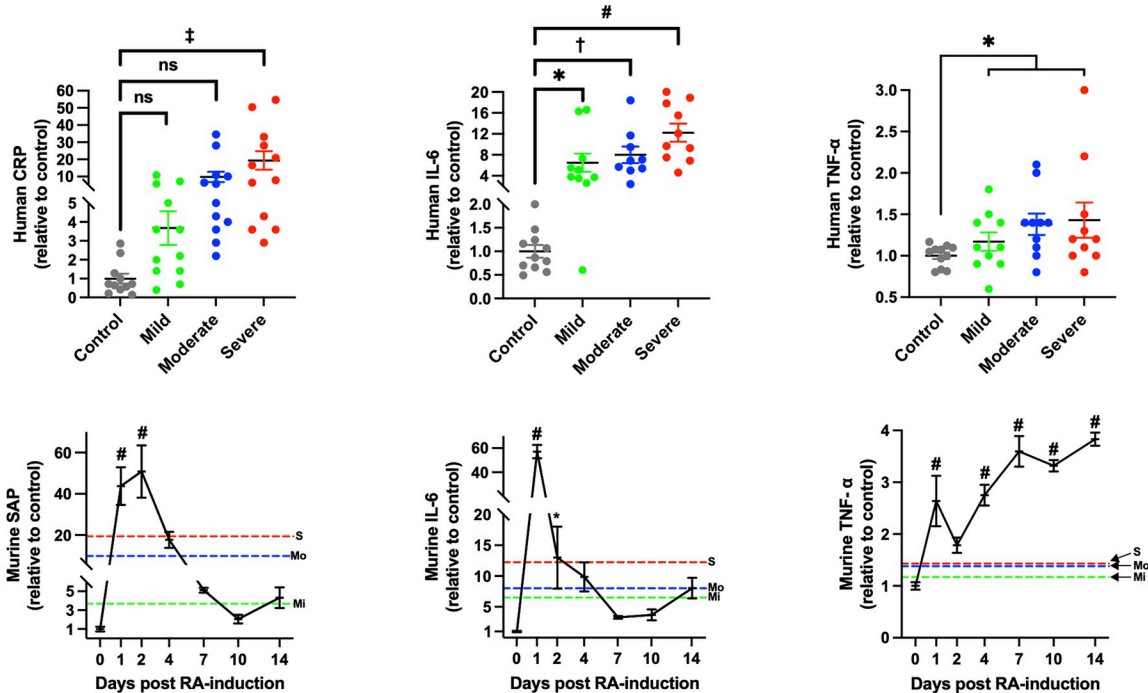

**Fig 2. Analysis and comparison of systemic inflammation in the DTHA model and RA patients.** *Panel A-C*: Serum levels of CRP (panel A), IL-6 (panel B) and TNF-α (panel C) from RA patients. Patients were chosen according to the DAS to represent three disease-states of severity (mild, moderate and severe). Each group consisted of 12 patients and age- and sex-matched healthy volunteers (control). CRP was analysed individually, and included 12 individuals in each group. One outlier was excluded in the healthy volunteer group. Due to space limiting reasons (limited wells in the 96-well) in the analysis of IL-6 and TNF-α, 10 patients were randomized for analysis in each disease group. Of these we excluded one outlier of IL-6 in the moderate group, one outlier of TNF-α in the control group. In addition, we excluded one IL-6 value in the control group, even though it did not meet our exclusion criterion, due to that the same individual had outlying both CRP and TNF-α values. Data is displayed as relative to control and presented as scatter plots with mean ± SEM. Statistical significance was calculated using one-way ANOVA with Bonferroni's multiple comparisons test for CRP and IL-6, and unpaired two-tailed *t*-test of control vs intervention (mild, moderate and severe combined) for TNF-α. *Panel D-F*: Temporal profiles of serum levels of murine SAP (panel D), IL-6 (panel E) and TNF-α (panel F) in DTHA mice relative to control. n = 10 for control group, n = 5 for other time points. Of these, two outliers for both SAP and IL-6 (representing the same two mice) were excluded in the control groups. Even though it did not meet our exclusion criterion, we chose to also exclude the corresponding TNF-α values from these two mice. Red grid line in graphs represents corresponding mean relative human value in severe human RA patients, blue grid line in graphs represents corresponding mean relative human value in moderate human RA patients, green grid line in graphs represents corresponding mean relative human value in mild human RA patients. Data is shown as mean ± SEM. S = severe, Mo = moderate, Mi = mild. Statistical significance was calculated in mice, relative to control, using one-way ANOVA with Bonferroni's multiple comparisons test. ns = non-significant, *$P < 0.05$, †$P < 0.01$; ‡$P < 0.001$, #$P < 0.0001$. DTHA, delayed type hypersensitivity arthritis; CRP, C-reactive protein; IL-6, interleukin 6; TNF-α, tumor necrosis factor α; DAS, Disease Activity Score; SAP, serum amyloid P component.

We determined the relative alterations of important inflammatory markers and mediators of RA in the DTHA model. As seen in Fig 2, panel D-F, both SAP (the murine counterpart of CRP), IL-6 and TNF-α were significantly elevated after induction of arthritis. To assess the clinical relevance of these alterations in the DTHA model, we analysed the human counterparts of these inflammatory mediators in serum from untreated RA patients and healthy controls. The details regarding patient characteristics have previously been published [23, 24]. We prespecified three groups of disease severity according to the DAS and within these groups we randomly selected patients for further analysis, keeping the male-to-female ratio at 1. Patient demographic data is shown in S1 Table. Fig 2 shows the increases of CRP (panel A), IL-6 (panel B) and TNF-α (panel C) in the three disease severity groups, relative to control. Graphs displaying absolute values are shown in S2 Fig. As expected, when analysing a relatively small

number of patient samples, variance is high, but with a clear trend for increased values for all measured markers. For IL-6 levels, all the three disease severity groups were significantly higher than control. Furthermore, when combining the three disease severity groups, levels of TNF-α and CRP were also significantly higher than the controls. Between the disease severity groups, only CRP and IL-6 were significantly different between mild and severe human RA. We did not statistically compare the human and mice results, as this would be futile due to the large variance in the data. However, we have graphically displayed the human average values (mild disease: green line, moderate disease: blue line and severe disease: red line) together with the mouse data (in Fig 2, panel D, E and F). Although not statistically compared, we believe the data overall supports the notion that our experimental DTHA model gives a clinically relevant humoral inflammation within the time span of ORAB-induced HF development.

## 3.2 HF parameters in the ORAB model

Cardiac dysfunction and heart failure was induced by constriction of the ascending thoracic aorta by a modification of previously established methodology. This refined technique allowed us to generate highly predictable cardiac phenotypes with low variance, as described in the method section [25]. Two different o-ring diameters were chosen to constrict the ascending aorta leading to cardiac stress, causing two different trajectories of cardiac dysfunction and HF [25]. There was highly significant step-wise increases in total heart weight and LV weight between sham-operated mice, ORAB 0.66 and ORAB 0.55 (Fig 3, panel B and C). These findings were corroborated by MRI where estimated LV mass for ORAB 0.55 and ORAB 0.66 were significantly higher than sham-operated mice at all post-surgery time points studied ($P < 0.0001$ for all comparisons made). Furthermore, the severity of aortic constriction afforded by ORAB 0.66 and ORAB 0.55 generated two distinct trajectories of cardiac hypertrophy and cardiac dysfunction as shown in Fig 4A and 4B. More interestingly, the two chosen degrees of aortic constriction yielded a binary result as to pulmonary congestion (Fig 3, panel D), as only ORAB 0.55 displayed increased lung weight.

## 3.3 Assessment of cardiac function after ORAB surgery

There was a steady, significant decline in systolic and diastolic function over time for both ORAB 0.55 and ORAB 0.66, as compared to sham-operated mice. These data were substantiated by measurement of LVEF (Fig 4, panel A), GLS (%, Fig 5, panel A), peak systolic strain rate (SRs', Fig 5, panel C), and peak early diastolic strain rate (SRe', Fig 5, panel B). All these measurements were showing a significant difference between sham and the two ORAB groups, with $P <0.0001$ for all comparisons made. Between the ORAB groups there was a significant difference in LVEF, SRs' and GLS between 0.66 vs 0.55 from the first MRI at week 2 ($P = 0.015$ for LVEF, $P < 0.0001$ for SRs' and GLS, respectively). These observations were substantiated by findings of mirroring foetal gene alterations (Fig 6, panel A-D). All four foetal gene expression levels were significantly elevated after ORAB surgery as compared to sham, but the data also substantiate the evidence for the step-like difference in HF development previously described, as mRNA levels of *NPPA* and *MYH7* were significantly higher in ORAB 0.55 compared to ORAB 0.66 (Fig 6, panel A and C).

## 3.4 Impact of arthritis on HF development

As previously described, we believe our model of RA represents relevant systemic inflammation in the context of HF. Furthermore, we reproduce that the ORAB model yields HF with graded clinical cardiac trajectories with clear evidence of both systolic and diastolic dysfunction. Still, when combining the models, no overall difference in HF development could be seen

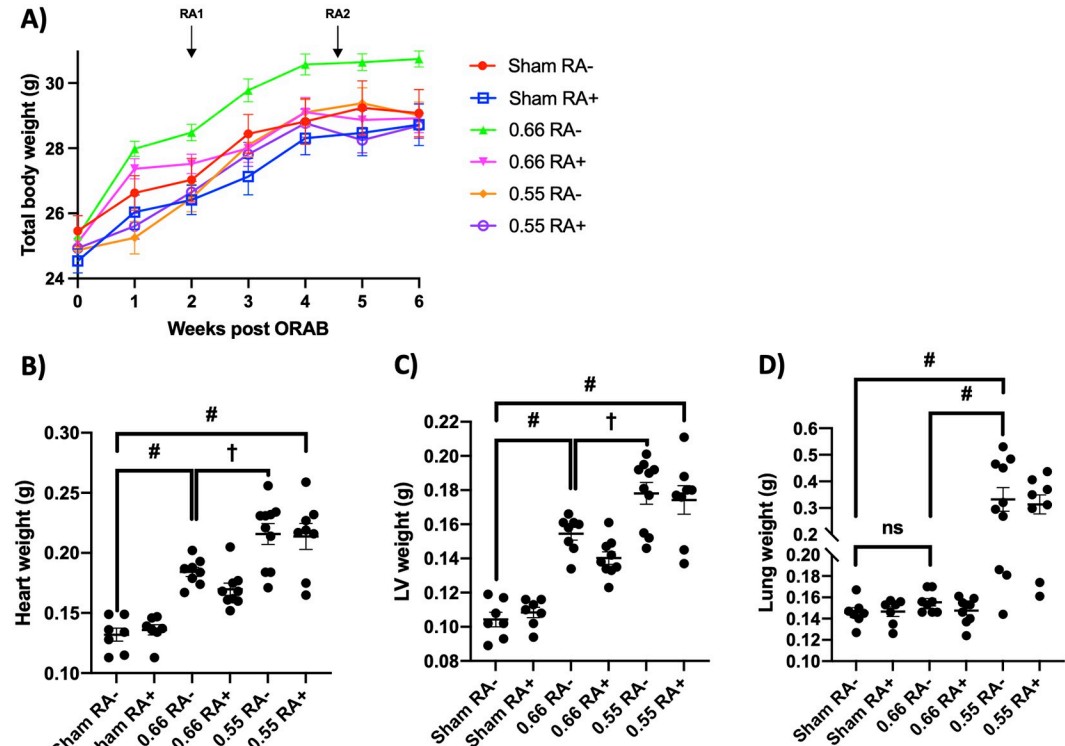

**Fig 3. Temporal body weight alterations and organ weights.** *Panel A*: Development of mice body weight within the various intervention groups throughout the timespan of the study (n = 7, 7, 8, 9, 10, 8 for sham RA-, sham RA+, ORAB 0.66 RA-, ORAB 0.66 RA+, ORAB 0.55 RA- and ORAB 0.55 RA+, respectively). Day 0 represents data before ORAB surgery, arthritis was induced at 2 weeks (RA1). At day 18 after RA induction, arthritis was re-established with an additional injection with mBSA (RA2). Data is mean ± SEM. *Panel B-D*: Total heart weight, left ventricular weight, and lung wet weight at the time of euthanization (n = 7, 7, 8, 9, 10, 8 for sham RA-, sham RA+, ORAB 0.66 RA-, ORAB 0.66 RA+, ORAB 0.55 RA- and ORAB 0.55 RA+, respectively). Data is presented as scatter plots with mean ± SEM. Statistical significance was calculated using one-way ANOVA with Bonferroni's multiple comparisons test. ns = non-significant, $^*P < 0.05$, $^†P < 0.01$; $^‡P < 0.001$, $^#P < 0.0001$. ORAB, o-ring aortic banding; mBSA, methylated bovine serum albumin.

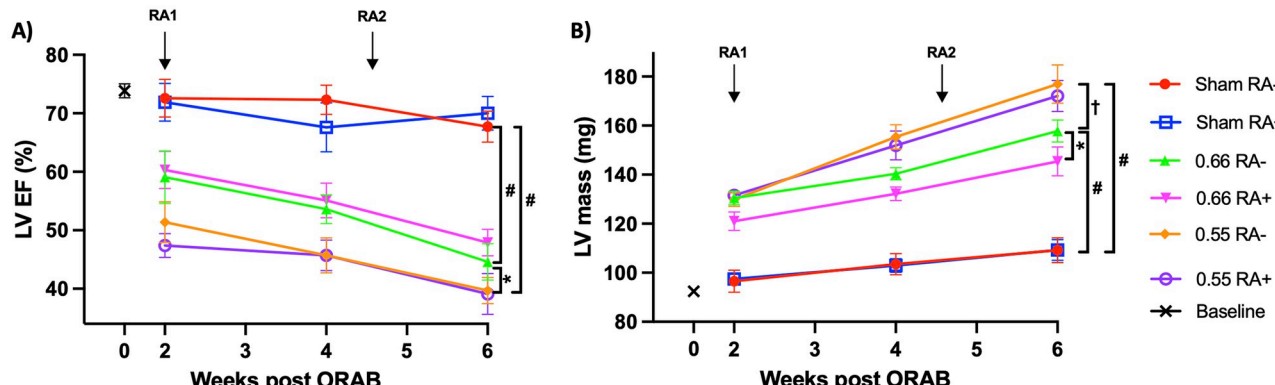

**Fig 4. Analysis of LV ejection fraction and LV myocardial mass.** MRI assessed LV ejection fraction (EF %; panel A) and LV myocardial mass (panel B) at given time points (n = 42 in total, 7 in each group). Data is presented as mean ± SEM. RA1 = time of establishment of RA phenotype with a sc. injection of mBSA, RA2 = time of re-establishment of RA phenotype with an additional injection with mBSA. Statistical significance was calculated using two-way ANOVA with Bonferroni's multiple comparisons test. $^*P < 0.05$, $^†P < 0.01$; $^‡P < 0.001$, $^#P < 0.0001$. MRI, magnetic resonance imaging; LV, left ventricular; mBSA, methylated bovine serum albumin.

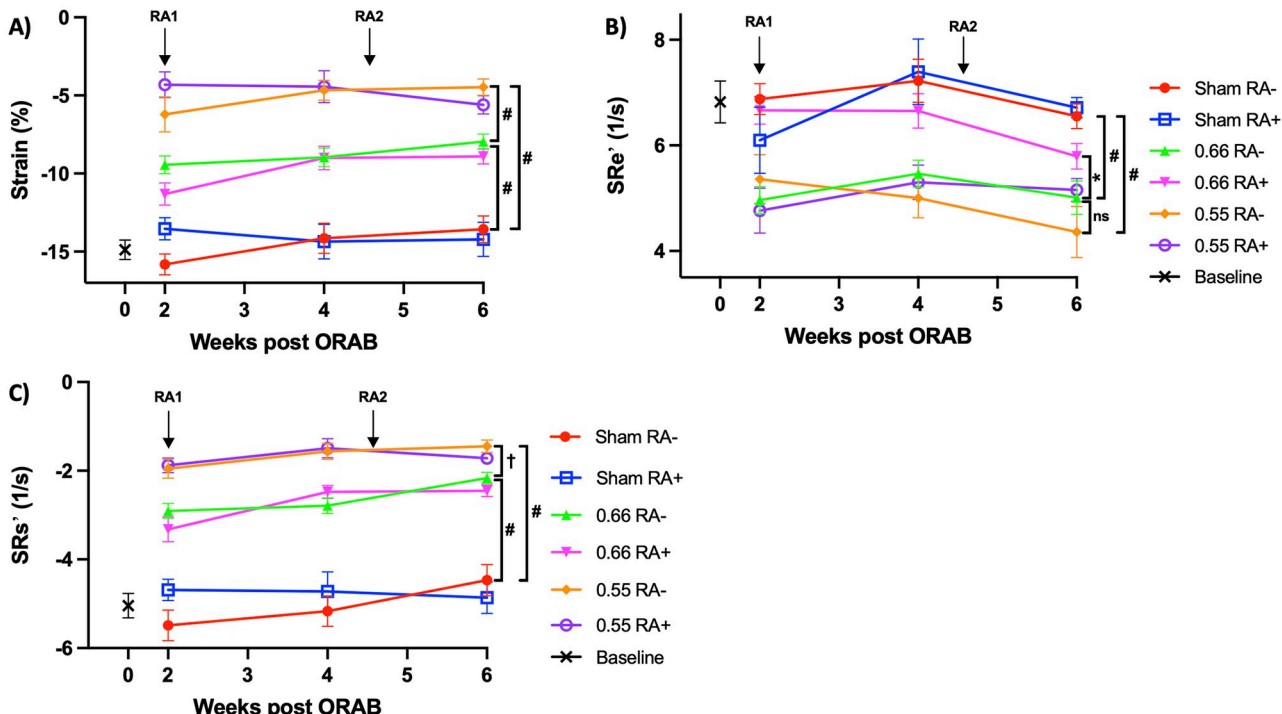

**Fig 5. Analysis of heart function parameters by TPM-MRI.** MRI assessed LV strain (%, panel A), strain rate e' (panel B; SRe', peak early diastolic strain rate) and strain rate s' (panel C; SRs', peak systolic strain rate) in septum and LV free wall combined at given time points (n = 42 in total, 7 in each group). Values are mean ± SEM. RA1 = time of establishment of RA phenotype with a sc. injection of mBSA, RA2 = time of reestablishment of RA with an additional injection with mBSA. Statistical significance was calculated using two-way ANOVA with Bonferroni's multiple comparisons test. ns = non-significant, $^*P < 0.05$, $^†P < 0.01$; $^‡P < 0.001$, #$P < 0.0001$. MRI, magnetic resonance imaging; LV, left ventricular; mBSA, methylated bovine serum albumin.

between mice with and without arthritis. The only numerical significant difference between arthritis and control was seen in a few isolated data sets in the ORAB 0.66 group (MRI assessed LV myocardial mass, SRe' and total body weight). Although this could leave the impression of an inflammatory beneficial effect on mild HF, it is important to emphasize that these

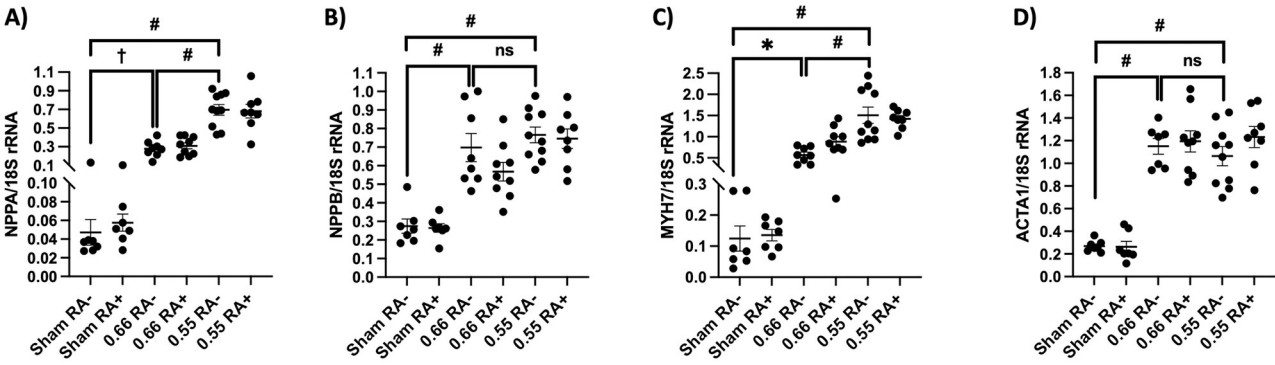

**Fig 6. Foetal gene activation in HF with or without arthritis.** Myocardial gene expression levels of *NPPA* (panel A), *NPPB* (panel B), *MYH7* (panel C) and *ACTA1* (panel D) was analysed at termination of the experimental study, relative to the housekeeping gene 18S (n = 7, 7, 8, 9, 10, 8 for sham RA-, sham RA+, ORAB 0.66 RA-, ORAB 0.66 RA+, ORAB 0.55 RA- and ORAB 0.55 RA+, respectively). One outlier was excluded in the analysis of *ACTA1* in the 0,66 RA- group. Data is displayed as scatter plots with mean ± SEM. Statistical significance was calculated using one-way ANOVA with Bonferroni's multiple comparisons test. ns = non-significant, $^*P < 0.05$, $^†P < 0.01$; $^‡P < 0.001$, #$P < 0.0001$. *NPPA*, natriuretic peptide A; *NPPB*, natriuretic peptide B; *MYH7*, myosin heavy chain 7; *ACTA1*, actin alpha 1, skeletal muscle.

differences appeared before induction of DTHA. Furthermore, similar differences could not be found in the more severe ORAB group, nor in the sham group. Thus, we conclude that these differences most likely is not due to arthritis. All the other data including MRI (Figs 4 and 5), foetal gene expression levels (Fig 6) and organ weights (Fig 3) provide *in sum* multiple lines of evidence that in this setting, the chosen experimental model of RA does not impact the development of HF in mice.

## 4. Discussion

RA is associated with HF [7, 9, 12, 30, 31] and data suggests there are RA-specific HF risk factors [7]. These risk factors correlate to RA disease activity [9] and are probably mediated through unknown aspects of systemic inflammation. Currently, it is unclear whether isolated RA is a sufficient promoter of HF, or whether RA needs to act in concert with an already HF-prone heart in order for HF to develop. Thus, the mechanisms underlying the association between RA and HF are unknown. We believe gaining such *in depth* insights will be challenging in human trials. On this note, although imperfect, experimental animal studies may provide data of potential clinical relevance on these questions. Such knowledge will be important when choosing RA treatments and in the development of specific treatments for HF within the RA/HF comorbidity, as well as identifying RA patients at risk of developing HF.

The main finding of the current study is that DTHA-mediated arthritis did not cause, nor influence the evolvement of HF in the ORAB HF model. We temporally assessed alterations of cardiac dimensions and function, alterations of organ weights and activation of the cardiac foetal gene program. All displayed clear differences between the two HF groups, as well as between the HF groups and sham control, but were not influenced by the presence of arthritis. Cardiac dysfunction/HF in RA has convincingly been demonstrated clinically, as well as in three different mouse RA models [15, 17, 18] and one rat RA model [16, 19]. Thus, the pivotal question in the current study is why we did not observe the same. There are several experimental parameters in the current study that could explain why DTHA-mediated arthritis did not impact cardiac function, ranging from type, degree and duration of systemic inflammation, type and duration of HF, sequence of disease introduction (HF vs arthritis), as well as mouse strain and sex.

Firstly, we believe the current study's experimental models of HF and RA are relevant of human disease [25, 26], and that combining these models is a valid way of observing the impact of arthritis-induced systemic inflammation on the natural evolvement of HF. By standardised constriction of the ascending aorta by different sized o-rings, cardiac dysfunction/HF was induced with a phenotype resembling in mechanism that of aortic stenosis and hypertension. It is important to emphasize that this HF model produces predictable cardiac pathological trajectories with a very low variation. As is clear in the current study, our chosen degrees of aortic constriction generated distinct paths of HF development, with clear and evolving cardiovascular dysfunction, but importantly with near binary outcomes in resulting cardiac decompensation and backward failure.

No ideal experimental model of RA exists, as human RA is considered a heterogenous disease with varying aetiology [32, 33] and there are major knowledge gaps regarding the pathogenesis [11, 14]. Common traits are considered to be the generation of neoantigens, which in turn activates both the innate and adaptive immune system and subsequent generation of arthrogenic autoantibodies [34, 35]. In such, the DTHA model mimics some aspects of the RA pathogenesis, as mBSA may represent the neoantigen and injecting anti-collagen II antibodies may mimic the generation of autoantibodies. Previous investigations have also demonstrated that arthritis in the DTHA model is treatable with clinically used biological disease modifying

anti rheumatic drugs [29]. However, and as previously commented on, we believe that any systemic effect of DTHA-mediated local arthritis would not be a consequence of autoimmunity, this because the arthritis phenotype observed is exclusively restricted to the mBSA-injected hind limb. Therefore, and of putative importance for interpretating the current study, any observed cardiac effects in the DTHA model would most likely be a consequence of systemic "spillover" of locally generated inflammation.

Thus, the most important aspect of the clinical validity of the current study, we believe to be the degree and duration of systemic inflammation in the DTHA model. To quantify systemic inflammation, we conducted a time study of DTHA and analysed levels of important markers and mediators of inflammation. To place these experimental data in clinical context, we analysed serum from untreated RA patients categorized with mild, moderate or severe disease according to a globally endorsed Disease Activity Score (DAS) and from healthy controls. Variation in the two data sets (murine vs human), and especially in the patient material, did not allow for adequate statistical comparison between murine and patient inflammatory responses. Such comparison could have provided some cues as to the clinical value of the DTHA model (i.e., the time-amount mice with HF were exposed to "clinical relevant inflammation"). However, as should be evident when studying the data as a whole, the DTHA model does indeed elicit systemic inflammation that both spatially and temporally appears clinically relevant, but albeit may still be of inadequate strength or duration.

This is the first experimental study where putative cardiac effects of DTHA-induced arthritis has been studied. Although few, some previous investigations on both mice and rats using three different models of experimental RA have demonstrated cardiac dysfunction [15–19]. Both diastolic dysfunction [16, 17, 19] and systolic dysfunction [15, 18] have been demonstrated. These studies share some common features that may explain the apparent difference in findings, i.e., their observed negative cardiac effects towards our lack of such. One obvious difference is that these studies, in contrast to our study, all display systemic, antibody-mediated polyarthritis. This could imply a common pathogenesis of arthritis and cardiac dysfunction, this being autoantibody-mediated myocardial damage. Although not excluded, we believe such a hypothesis is less likely as the antibodies involved in these experimental RA-models are very different in generation and action. Clinical data showing an association between very different autoimmune diseases and HF also support the notion of another common denominator than autoantibodies [20–22].

As to our knowledge, none of the above studies have examined the duration and magnitude of systemic inflammation elicited by their RA-models. As argued for above, we believe the DTHA-induced systemic inflammation to be clinically relevant. Even so, the other RA-models may elicit a more robust and durable systemic inflammation due to their polyarthritic nature, as well as visually being of more chronic nature. In fact, Mokotedi et al did find a correlation between levels of inflammatory mediators and degree of cardiac dysfunction in the collagen-induced arthritis rat model [16]. The latter harmonizing with clinical observations of a correlation between RA disease degree and cardiac dysfunction [9, 12, 13]. We ended our study at day 28 after induction of arthritis and some of the other studies have had longer observational periods [15–17]. Thus, even though we believe it unlikely, we cannot exclude that extending our observation period would reveal differences in cardiac dysfunction.

Finally, the risk of cardiac dysfunction/HF amongst RA patients probably requires a genetic predisposition. In studies where cardiac function in experimental mouse RA-models has been studied, the strain has not been C57BL/6J (which was used in the current study). Thus, we cannot exclude the possibility that such genetic differences, in this case C57BL/6J vs BALBc or DBA/Ij, may play a role in the susceptibility for cardiac dysfunction in systemic inflammation.

To conclude, even though we believe that the DTHA model is a clinically relevant model of systemic inflammation in RA, it did not induce cardiac dysfunction, nor alter the course of HF induced by aortic constriction, within our set experimental boundaries. In context with the other experimental studies that have found cardiac dysfunction in RA, the current study provides a foundation for further studies exploring the requirements needed for RA to induce cardiac dysfunction/HF.

## Supporting information

**S1 Fig. Graphical abstract.** Schematic timeline of the experimental procedures.
(TIF)

**S2 Fig. Analysis and comparison of systemic inflammation in the DTHA model and RA patients (absolute values).** *Panel A-C*: Serum levels of CRP (panel A), IL-6 (panel B) and TNF-α (panel C) from RA patients. Patients were chosen according to the DAS to represent three disease-states of severity (mild, moderate and severe). Each group consisted of 12 patients and age- and sex-matched healthy volunteers (control). CRP was analysed individually, and included 12 individuals in each group. One outlier was excluded in the healthy volunteer group. Due to space limiting reasons (limited wells in the 96-well) in the analysis of IL-6 and TNF-α, 10 patients were randomized for analysis in each disease group. Of these we excluded one outlier of IL-6 in the moderate group, one outlier of TNF-α in the control group, and one outlier of CRP in the control group. In addition, we excluded one IL-6 value in the control group, even though it did not meet our exclusion criterion, due to that the same individual had outlying both CRP and TNF-α values. Data is displayed as absolute values and presented as scatter plots with mean ± SEM. Statistical significance was calculated using one-way ANOVA with Bonferroni's multiple comparisons test for CRP and IL-6, and unpaired two-tailed *t*-test of control vs intervention (mild, moderate and severe combined) for TNF-α. *Panel D-F*: Temporal profiles of serum levels of murine SAP (panel D), IL-6 (panel E) and TNF-α (panel F) in DTHA mice. Absolute values are shown. n = 10 for control group, n = 5 for other time points. Of these, two outliers for both SAP and IL-6 (representing the same two mice) were excluded in the control groups. Even though it did not meet our exclusion criterion, we chose to also exclude the corresponding TNF-α values from these two mice. Data is shown as scatter plots with mean ± SEM. Statistical significance was calculated using one-way ANOVA with Bonferroni's multiple comparisons test, relative to control. ns = non-significant, $^{*}P < 0.05$, $^{†}P < 0.01$; $^{‡}P < 0.001$, #$P < 0.0001$. DTHA, delayed type hypersensitivity arthritis; CRP, C-reactive protein; IL-6, interleukin 6; TNF-α, tumor necrosis factor α; DAS, Disease Activity Score; SAP, serum amyloid P component.
(TIF)

**S1 Table. Baseline characteristics.** Baseline characteristics of patients and healthy controls in the supplemental study. Data is presented as mean values with 95% confidence interval. DAS, Disease Activity Score; SJC-44, swollen joint counts among 44 joints; CRP, C-reactive protein; SR, sedimentation rate.
(TIF)

**S1 File. Dataset.**
(XLSX)

## Acknowledgments

The authors wish to thank Lili Zhang and Merete Høyem for expert technical assistance. Grethe-Elisabeth Stenvik is thanked for assistance in the selection and collection of serum samples from RA patients, and Liv Wenche Thorbjørnsen is thanked for assistance in the collection of serum samples from the healthy controls.

## Author Contributions

**Conceptualization:** Håvard Attramadal, Ivar Sjaastad, Leif Erik Vinge.

**Data curation:** Theis Christian Tønnessen, Ida Marie Hauge-Iversen, Emil Knut Stenersen Espe, Mohammed Shakil Ahmed, Thor Ueland, Håvard Attramadal, Ivar Sjaastad, Leif Erik Vinge.

**Formal analysis:** Theis Christian Tønnessen, Ida Marie Hauge-Iversen, Emil Knut Stenersen Espe, Mohammed Shakil Ahmed, Thor Ueland, Leif Erik Vinge.

**Funding acquisition:** Håvard Attramadal, Leif Erik Vinge.

**Investigation:** Theis Christian Tønnessen, Leif Erik Vinge.

**Methodology:** Arne Olav Melleby, Ida Marie Hauge-Iversen, Emil Knut Stenersen Espe, Sara Marie Atkinson, Ivar Sjaastad, Leif Erik Vinge.

**Project administration:** Theis Christian Tønnessen, Håvard Attramadal, Ivar Sjaastad, Leif Erik Vinge.

**Resources:** Espen Andre Haavardsholm, Espen Melum, Håvard Attramadal, Ivar Sjaastad, Leif Erik Vinge.

**Software:** Ivar Sjaastad.

**Supervision:** Håvard Attramadal, Ivar Sjaastad, Leif Erik Vinge.

**Writing – original draft:** Theis Christian Tønnessen, Leif Erik Vinge.

**Writing – review & editing:** Theis Christian Tønnessen, Arne Olav Melleby, Ida Marie Hauge-Iversen, Emil Knut Stenersen Espe, Mohammed Shakil Ahmed, Thor Ueland, Espen Andre Haavardsholm, Sara Marie Atkinson, Espen Melum, Håvard Attramadal, Ivar Sjaastad, Leif Erik Vinge.

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
