## [Decision Letter · Decision Letter 0]

8 Oct 2021

PONE-D-21-27012Impact of experimental Rheumatoid Arthritis on development of Heart FailurePLOS ONE

Dear Dr. Tønnessen,

Thank you for submitting your manuscript to PLOS ONE. After careful consideration, we feel that it has merit but does not fully meet PLOS ONE’s publication criteria as it currently stands. Therefore, we invite you to submit a revised version of the manuscript that addresses the points raised during the review process.

We look forward to receiving your revised manuscript.

Kind regards,

Michael Bader

Academic Editor

PLOS ONE

Journal Requirements:

2. Please modify the title to ensure that it is meeting PLOS’ guidelines (https://journals.plos.org/plosone/s/submission-guidelines#loc-title). In particular, the title should be "specific, descriptive, concise, and comprehensible to readers outside the field" and in this case we feel it is not informative and specific about your study's scope and methodology.

Reviewers' comments:

Reviewer's Responses to Questions

**Comments to the Author**

1. Is the manuscript technically sound, and do the data support the conclusions?

Reviewer #1: Partly

Reviewer #2: Yes

2. Has the statistical analysis been performed appropriately and rigorously? 

Reviewer #1: Yes

Reviewer #2: No

3. Have the authors made all data underlying the findings in their manuscript fully available?

Reviewer #1: Yes

Reviewer #2: Yes

4. Is the manuscript presented in an intelligible fashion and written in standard English?

Reviewer #1: Yes

Reviewer #2: Yes

5. Review Comments to the Author

Reviewer #1: This manuscript reports an interesting study aiming to address a highly relevant topic; the effect of RA on the development of heart failure. Unfortunately, there are a few concerns and publication at this stage seems premature.

Major comments:

+ Although not many, there are studies on cardiac effects in animal models of RA, especially CIA (e.g. Reynolds et al, 2012; He et al 2013; Palma et al 2016, reviewed in Sanghera et al 2019), which should be discussed.

+ Structure/reporting: The value of this studies clearly lies in the fact that this model of DTH-RA has been tested in conjunction with HF. There is obvious value in reporting the findings for other researchers who might want to test a similar hypothesis, but the study should then be described accordingly with an emphasis on the models, rather than overselling relevance and impact of a potential difference that is not present. The manuscript in its present form focuses too much on a potential difference in outcome +/-RA which has not been observed.

+ The choice of experimental design is surprising. Systemic autoimmune diseases underly cardiac complications and may exacerbate development of heart disease. While this is well established for a range of autoimmune conditions, the clinical relevance of pressure-induced HF followed by autoimmunity is questionable. If the authors aim to address the clinical problem they refer to, autoimmunity needs to be induced before heart failure not after.

+ The model used here is delayed-type hypersensitivity (which is not autoimmunity) with auto-Abs targeting an individual antigen (collagen). While no model of systemic autoimmunity is perfect, there are well accepted RA models that reflect human pathology better. Some of them can be time consuming and rather tedious to induce, so it is much appreciated that alternatives are being sought, but this needs to be reflected in the description of the study, starting with a precise title.

Minor comments:

+ It’s unclear why well established patient data should be repeated in a suboptimal sample size. Why not reference previous reports?

+ There are a few instances where ‘outliers have been excluded’. It is unclear what criteria were used to define outliers and how exclusion has been justified.

+ There appears to be a small beneficial effect or the DTH-RA model on mild HF in some of the readouts? This might warrant discussion and further investigation considering that ‘fresh’ inflammation might in fact be beneficial if introduced at the right time.

Reviewer #2: The authors have sought to investigate the link between RA and heart failure by assessing the impact of RA on cardiac function using a murine model of thoracic aortic constriction. As they indicate, RA patients are at increased risk of HF and the mechanisms driving this are still to be defined. This is therefore an important study and the manuscript is well written.

Comments:

1. It is interesting that the authors have chosen to study the effect of arthritis on the development of HF rather than the effect of arthritis on the development of cardiac dysfunction, although the rationale for this is explained. They have chosen an aortic constriction model driven by increased LV afterload. In the introduction both HFpEF and HFrEF is discussed and the literature suggests that RA patients may be particularly susceptible to HFpEF, which is driven by systemic inflammation (PMID: 34523821). However, the model chosen by the authors results in impaired EF. The authors are correct that there are few animal models to investigate HF in the context of RA and indeed models of diastolic dysfunction or HFpEF although there a few more examples in the literature that the authors may wish to discuss as the Pironti study is not the only experimental study that has investigated the cardiac consequence of RA as claimed in the discussion (PMID: 32208438; PMID: 33427610; PMID: 34526398). The authors might want to add a few lines to the discussion to distinguish between the different models of cardiac dysfunction and their relevance to the RA community.

2. A timeline for the experimental protocol would be useful indicating when ORAB was initiated and then when RA was induced.

3. Figure 1 panel A is this data mean +/- SEM (n=25) as indicated for panel B? Please add to panel A legend. No need to indicate ORAB in this legend as this data is not shown here.

4. Figure 2 (panel A-C) I am not sure what the authors mean by ‘due to space limiting reasons in the analysis……, 10 patients were randomised for analysis in each disease group’, yet there are 11 data points per group and it states that one outlier has been removed from each group. It is also not clear to me looking at the absolute values in supplemental figure 1, which patient(s) have been excluded as outliers as well. Also, why show the data as relative to control and then have another figure with the absolute values, rather than just showing absolute values in the main manuscript. There is some variance, as the authors have stated, but this is to be expected with human samples.

5. Were cytokine levels assessed in mice after the re-establishment of RA at days 15/18?

6. Panel 2 (D-F) as above, I am not sure why 2 data points have been excluded for SAP and IL-6 and not TNF and again why data is shown as relative to control as well as absolute data in the supplemental file. Is this because statistical significance was not reached in the murine data when using absolute values? It states in the text, lines 303-306 that ‘Although non-significant due to variance in the data, we believe the data overall supports the notion that our experimental DTHA model…’ This leads me to believe that the absolute data shown in supplemental figure 2 is being referred to here, not the relative data shown in figure 2 where significance is shown. I agree with the author’s interpretation of the data and would just show absolute values as shown in the supplemental figure. I do not think you can correlate the relative data in mice to levels in patients, or that it is necessary to do so.

7. Line 303: It is not clear here that the text is now describing the murine data. Line 304 states CRP, rather than SAP.

8. Lines 407-411 in the discussion are not clear to me. I am not sure what the authors mean by time-amount mice with HF were exposed to ‘clinical relevant inflammation’.

Minor comment

Yellow lines are difficult to see clearly in the figures.

Legend for figure 3 panel B-D ends ‘mBSA, methylated’. Sentence is incomplete, but mBSA is not shown in the figure so this can be deleted – also in the legends for figures 4 and 5.

6. PLOS authors have the option to publish the peer review history of their article (what does this mean?). If published, this will include your full peer review and any attached files.

Reviewer #1: No

Reviewer #2: No

---

## [Author Response · Author response to Decision Letter 0]

2 Dec 2021

Reviewers' comments:

Reviewer #1: This manuscript reports an interesting study aiming to address a highly relevant topic; the effect of RA on the development of heart failure. Unfortunately, there are a few concerns and publication at this stage seems premature.

Major comments:

+ Although not many, there are studies on cardiac effects in animal models of RA, especially CIA (e.g. Reynolds et al, 2012; He et al 2013; Palma et al 2016, reviewed in Sanghera et al 2019), which should be discussed.

+ Structure/reporting: The value of this studies clearly lies in the fact that this model of DTH-RA has been tested in conjunction with HF. There is obvious value in reporting the findings for other researchers who might want to test a similar hypothesis, but the study should then be described accordingly with an emphasis on the models, rather than overselling relevance and impact of a potential difference that is not present. The manuscript in its present form focuses too much on a potential difference in outcome +/-RA which has not been observed.

Answer: We do appreciate these comments, comments also raised by reviewer #2. We believe that addressing these comments have significantly elevated the quality of the paper. The findings in these studies against the (lack of) findings in the current study have now been discussed in detail, with emphasis on putative explanations for the different results. The current manuscript has been thoroughly revised positioning our findings in context of previous investigations, and in particular problematized different aspects of our choice of experimental models and setup. In such, we hope the current manuscript appear more nuanced.

+ The choice of experimental design is surprising. Systemic autoimmune diseases underly cardiac complications and may exacerbate development of heart disease. While this is well established for a range of autoimmune conditions, the clinical relevance of pressure-induced HF followed by autoimmunity is questionable. If the authors aim to address the clinical problem they refer to, autoimmunity needs to be induced before heart failure not after.

Answer: In hindsight we agree on this comment, and can assure the reviewer that this will be addressed in the future design of new experimental studies. The reason for the current experimental design was that we wanted to avoid the putative effects of arthritis-induced systemic inflammation on immediate peri- and postsurgical (cardiac) recovery. Also, as we believe an important question as to the vulnerability for RA patients to develop cardiac dysfunction/HF may be an already existing (subclinical) cardiac dysfunction, the current design would be of value.

+ The model used here is delayed-type hypersensitivity (which is not autoimmunity) with auto-Abs targeting an individual antigen (collagen). While no model of systemic autoimmunity is perfect, there are well accepted RA models that reflect human pathology better. Some of them can be time consuming and rather tedious to induce, so it is much appreciated that alternatives are being sought, but this needs to be reflected in the description of the study, starting with a precise title.

Answer: We agree that there are other models (like the CIA model) that may mimic its human counterpart better, especially in respect to autoimmunity and generalized arthritis (even though the specific autoimmunity in the human condition is quite different). Therefore, the putative value of the DTHA model we believe is that it may reflect the indirect consequence of arthritis-induced systemic inflammation. The differences in the various arthritis models are of importance and have now been addressed throughout the revised manuscript, starting with the title.

Minor comments:

+ It’s unclear why well established patient data should be repeated in a suboptimal sample size. Why not reference previous reports?

Answer: Our main objective in the analysis of the human RA serum was to relatively put into context the inflammatory responses seen in the DTHA model. As the ARCTIC study initially did not include age- and sex matched healthy controls, we needed to reanalyse a representative, though smaller subset of the ARCTIC study now including such controls. 

Clarifications on this matter have been added to the text: Page 14, line 273-277 (in revised manuscript with track changes).

+ There are a few instances where ‘outliers have been excluded’. It is unclear what criteria were used to define outliers and how exclusion has been justified.

Answer: Outliers were excluded according to the ROUT method with a Q value of 1% (Q is the maximum desired false discovery rate). Information of this is added to the method section 2.8. In addition, we excluded TNF-values from two control mice even though they did not satisfy our exclusion criterion. In these mice both IL-6 and SAP values were defined as outliers which led us to consider these mice, or sampling of these mice, as erroneous. We also excluded an IL-6 value from the human control group due to similar considerations (both CRP and TNF values being outliers). This was not specified in the original manuscript, and all the individual excluded values are now specified in the relevant figure texts (Fig 2 and 6, and S2 Fig). The exclusion of these murine TNF and human IL-6 values did not influence the statistical results in respect to significance. The complete data including absolute values of outliers can be viewed in the raw data file attached (S1_File_Dataset). 

+ There appears to be a small beneficial effect or the DTH-RA model on mild HF in some of the readouts? This might warrant discussion and further investigation considering that ‘fresh’ inflammation might in fact be beneficial if introduced at the right time.

Answer: We agree that some of the data comparing the impact of DTHA on cardiac function (i.e., SRe’) and weights (this including MRI estimated LV mass) leaves the impression of inflammatory beneficial effects on mild HF. However, in regards to function (Fig. 5B), this appears before inflammation is instigated. In respect to weights, an unexplainable difference in body weight (again appearing before induction of arthritis) may explain the differences. We now realize that the previous manuscript might have been unclear on this matter, and the current manuscript includes corrections in page 23, line 422-426 (in revised manuscript with track changes).

Reviewer #2: The authors have sought to investigate the link between RA and heart failure by assessing the impact of RA on cardiac function using a murine model of thoracic aortic constriction. As they indicate, RA patients are at increased risk of HF and the mechanisms driving this are still to be defined. This is therefore an important study and the manuscript is well written.

Comments:

1. It is interesting that the authors have chosen to study the effect of arthritis on the development of HF rather than the effect of arthritis on the development of cardiac dysfunction, although the rationale for this is explained. They have chosen an aortic constriction model driven by increased LV afterload. In the introduction both HFpEF and HFrEF is discussed and the literature suggests that RA patients may be particularly susceptible to HFpEF, which is driven by systemic inflammation (PMID: 34523821). However, the model chosen by the authors results in impaired EF. The authors are correct that there are few animal models to investigate HF in the context of RA and indeed models of diastolic dysfunction or HFpEF although there a few more examples in the literature that the authors may wish to discuss as the Pironti study is not the only experimental study that has investigated the cardiac consequence of RA as claimed in the discussion (PMID: 32208438; PMID: 33427610; PMID: 34526398). The authors might want to add a few lines to the discussion to distinguish between the different models of cardiac dysfunction and their relevance to the RA community.

Answer: We completely agree with the reviewer. These comments were also voiced by reviewer #1, and we refer to the answers given above. In short, these important studies have now been included and thoroughly discussed up against our data throughout the revised manuscript.

2. A timeline for the experimental protocol would be useful indicating when ORAB was initiated and then when RA was induced.

Answer: We agree that this would make the protocol clearer, and have uploaded a figure with a schematic illustration of the timeline in the supplemental material, see Figure S1.

3. Figure 1 panel A is this data mean +/- SEM (n=25) as indicated for panel B? Please add to panel A legend. No need to indicate ORAB in this legend as this data is not shown here.

Answer: This is both corrected in the manuscript.

4. Figure 2 (panel A-C) I am not sure what the authors mean by ‘due to space limiting reasons in the analysis……, 10 patients were randomised for analysis in each disease group’, yet there are 11 data points per group and it states that one outlier has been removed from each group. It is also not clear to me looking at the absolute values in supplemental figure 1, which patient(s) have been excluded as outliers as well.

Answer: From the databases we randomly selected 12 patients and controls. The demographic and some analytical data on these patients are all displayed in the supplemental table 1. Analysis of CRP was done separately, and only one of the datapoints were excluded as it met our exclusion criterion (in the control group). For the analysis of IL-6 and TNF we did have limitations in respect to the number of patients we could include in the 96-well plate. Thus, we chose to have 12 samples in the control group, and 10 samples in the disease groups (mild, moderate, severe). Of these, one outlier was removed in the control CRP group and one in the control TNF group. These outliers were from the same individual, and thus we chose also to exclude the corresponding IL-6 value. We also excluded one outlier in the moderate IL-6 group. We now realize that there is one discrepancy between the graphs displaying relative and absolute data (the excluded IL-6 datapoint in the control group). This has now been corrected (new absolute graph, S2 Fig). All the data, including the excluded data, can be seen in the raw data file attached (S1_File_Dataset).

The above considerations are added in detail in the now revised manuscript. See changes done in figure legend describing figure 2. Similar considerations/detailing have also been added for the murine data in the revised manuscript.

Also, why show the data as relative to control and then have another figure with the absolute values, rather than just showing absolute values in the main manuscript. There is some variance, as the authors have stated, but this is to be expected with human samples.

Answer: We appreciate the comments from reviewer #2. The purpose of this experiment was to compare inflammatory responses of mice subjected to experimental arthritis to human RA patients. We believe such comparisons between two different species are better visualized and represented using relative measures, as baseline values often differ (as they do here). We do, however, appreciate the importance of showing non-corrected data, and have thus included absolute presentations in the online supplement.

5. Were cytokine levels assessed in mice after the re-establishment of RA at days 15/18?

Answer: This is a valid question as it would add additional value to the interpretation of the DTHA model. We did consider to extend this part of our experiment, but chose not to. This decision was due to a (possibly erroneous) perception of minor additional informative value, but which would come with substantial additional financial costs and with an increase of mice needed.

6. Panel 2 (D-F) as above, I am not sure why 2 data points have been excluded for SAP and IL-6 and not TNF and again why data is shown as relative to control as well as absolute data in the supplemental file. Is this because statistical significance was not reached in the murine data when using absolute values? It states in the text, lines 303-306 that ‘Although non-significant due to variance in the data, we believe the data overall supports the notion that our experimental DTHA model…’ This leads me to believe that the absolute data shown in supplemental figure 2 is being referred to here, not the relative data shown in figure 2 where significance is shown. I agree with the author’s interpretation of the data and would just show absolute values as shown in the supplemental figure. I do not think you can correlate the relative data in mice to levels in patients, or that it is necessary to do so.

Answer: Regarding excluded datapoints, the rationale for these decisions has been thoroughly explained above (see also comments to reviewer #1). The relative and absolute analysis included the same values/datasets (with one error now corrected, see above) and the results are the same. However, we understand reviewer #2’s confusion as we have not displayed symbols of significance in the graph. This, and corresponding figure legend, has now been corrected.

Regarding the comparisons between the relative datasets of the inflammatory markers in mice and human, we now realize that our figure is confusing. The statistical annotation in Fig 2, D-F is not that of mouse vs human comparison. In order to visually appreciate the relative inflammatory responses on our arthritis model, we have inserted the average human values representing mild, moderate and severe disease. This we still believe is the most convenient way to display these data as the sole purpose of reanalyzing the clinical material, now including healthy volunteers, was to put the murine inflammatory responses into clinical context. We hope the altered text describing this (page 21, line 372-381 in revised manuscript with track changes) is more explanatory.

7. Line 303: It is not clear here that the text is now describing the murine data. Line 304 states CRP, rather than SAP.

Answer: We are describing the human data in line 303, which has now been specified and are hopefully now clearer (page 21, line 372 in revised manuscript with track changes).

8. Lines 407-411 in the discussion are not clear to me. I am not sure what the authors mean by time-amount mice with HF were exposed to ‘clinical relevant inflammation’.

Answer: We realize that this part of the discussion section is somewhat confusing and we have now tried to clarify it (see page 27, line 493-497 in revised manuscript with track changes).

Minor comment

Yellow lines are difficult to see clearly in the figures.

Answer: This has been changed to blue colour in fig 2, and pink colour in fig 3A, 4 and 5.

Legend for figure 3 panel B-D ends ‘mBSA, methylated’. Sentence is incomplete, but mBSA is not shown in the figure so this can be deleted – also in the legends for figures 4 and 5.

Answer: The reason for the incomplete sentence was a formatting error in the figure legend textbox. We chose to gather all the abbreviations at the end of the figure legend, and not at the end of each separate panel. Therefore, mBSA was explained after the figure legend of figure 3, panel B-D.

---

## [Decision Letter · Decision Letter 1]

6 Jan 2022

Impact of delayed type hypersensitivity arthritis on development of heart failure by aortic constriction in mice

PONE-D-21-27012R1

Dear Dr. Tønnessen,

We’re pleased to inform you that your manuscript has been judged scientifically suitable for publication and will be formally accepted for publication once it meets all outstanding technical requirements.

Kind regards,

Michael Bader

Academic Editor

PLOS ONE

Additional Editor Comments (optional):

Reviewers' comments:

Reviewer's Responses to Questions

**Comments to the Author**

1. If the authors have adequately addressed your comments raised in a previous round of review and you feel that this manuscript is now acceptable for publication, you may indicate that here to bypass the “Comments to the Author” section, enter your conflict of interest statement in the “Confidential to Editor” section, and submit your "Accept" recommendation.

Reviewer #1: All comments have been addressed

Reviewer #2: All comments have been addressed

2. Is the manuscript technically sound, and do the data support the conclusions?

Reviewer #1: Yes

Reviewer #2: Yes

3. Has the statistical analysis been performed appropriately and rigorously? 

Reviewer #1: Yes

Reviewer #2: Yes

4. Have the authors made all data underlying the findings in their manuscript fully available?

Reviewer #1: Yes

Reviewer #2: Yes

5. Is the manuscript presented in an intelligible fashion and written in standard English?

Reviewer #1: Yes

Reviewer #2: Yes

6. Review Comments to the Author

Reviewer #1: (No Response)

Reviewer #2: The manuscript is much clearer. The discussion is particular is much improved and puts the research in context.

7. PLOS authors have the option to publish the peer review history of their article (what does this mean?). If published, this will include your full peer review and any attached files.

Reviewer #1: No

Reviewer #2: No

---

## [Editor Report · Acceptance letter]

10 Jan 2022

PONE-D-21-27012R1 

Impact of delayed type hypersensitivity arthritis on development of heart failure by aortic constriction in mice 

Dear Dr. Tønnessen:

I'm pleased to inform you that your manuscript has been deemed suitable for publication in PLOS ONE. Congratulations! Your manuscript is now with our production department. 

Kind regards, 

on behalf of

Prof. Michael Bader 

Academic Editor

PLOS ONE